# Autoimmune Thyroid Disease is Associated with a Lower Prevalence of Diabetic Retinopathy in Patients with Type 1 Diabetic Mellitus

**DOI:** 10.3390/medicina56060255

**Published:** 2020-05-26

**Authors:** Magdalena Maria Stefanowicz-Rutkowska, Wojciech Matuszewski, Elżbieta Maria Bandurska-Stankiewicz

**Affiliations:** Clinic of Endocrinology, Diabetology and Internal Medicine, School of Medicine, Collegium Medicum, University of Warmia and Mazury in Olsztyn, 10-719 Olsztyn, Poland; wmatuszewski82@wp.pl (W.M.); bandurska.endo@gmail.com (E.M.B.-S.)

**Keywords:** diabetes, diabetic retinopathy, autoimmune thyroid disease

## Abstract

*Background and objectives*: The aim of the study was to assess the correlation of autoimmune thyroid diseases (AITD) in patients with diabetes mellitus type 1 (DM1) with the occurrence of diabetic retinopathy (DR). *Materials and Methods*: The inclusion criteria for the study were: type 1 diabetes diagnosed on the basis of WHO criteria lasting at least a year, presence of AITD for at least a year, and age over 18 years. The control group consisted of patients without diagnosed AITD (DM1noAITD), selected according to age, BMI and DM1 duration. Anthropometric parameters, metabolic risk factors such as glycated hemoglobin (HbA1c), lipids and blood pressure, thyroid status and the presence of DR were assessed. *Results*: The study involved 200 patients with type 1 diabetes aged 36 ± 12 years, 70 men and 130 women. Patients from the study group (DM1AITD) had significantly lower creatinine concentration, significantly lower systolic blood pressure (SBP), glycated hemoglobin (HbA1c) percentage and triglyceride (TG) concentration, and higher high-density lipoprotein (HDL-cholesterol) concentration than the control group (DM1noAITD). There was a significantly lower chance of non-proliferative diabetic retinopathy (NPDR) among DM1AITD than in the control group. *Conclusions*: Patients with DM1 and AITD were metabolically better balanced, as evidenced by a significantly lower SBP, percentage of HbA1c and TG, as well as significantly higher HDL-cholesterol in this group. Patients with DM1 and AITD were significantly less likely to have NPDR than the control group.

## 1. Introduction

Diabetes mellitus (DM) is a group of metabolic diseases characterized by hyperglycemia resulting from a defect in insulin secretion and/or action [1]. DM is the only non-communicable disease recognized by the WHO as an epidemic. In 1980, the number of DM patients in the world was 153 million, while in 2017 it amounted to 424 million. By 2045, the number of patients is predicted to reach 628 million [2]. Thyroid disease and diabetes mellitus (DM) are the two most common endocrinopathies occurring in clinical practice. DM and thyroid dysfunction exert a mutual influence. On one hand, thyroid hormones regulate carbohydrate metabolism and pancreas functions; on the other hand, DM affects both the function and the work of the thyroid gland [3,4]. Pathophysiological relations between thyroid diseases and DM are indicated by an increasing number of studies on their biochemical, genetic and hormonal interactions. The relationship between DM and thyroid disorders is characterized by a complex interaction [5]. Diabetic retinopathy (DR) is a highly specific microvascular complication of DM, and the final stage of its clinical manifestation is the loss of vision. DR is responsible for up to 80% of blindness among DM patients, constituting a significant cause of vision loss on a global scale [6,7,8,9]. There are few reports in the literature about the effect of thyroid disease on the development and course of DR in patients with DM. The primary objective of the study was to assess the correlation of autoimmune thyroid diseases (AITD) in patients with diabetes mellitus type 1 (DM1) with the presence of diabetic retinopathy (DR).

## 2. Materials and Methods 

### 2.1. A Retrospective, Cross-Sectional and Non-Interventional Study of Participants with DM1

A study was conducted in a patient population with diagnosed DM1 based on World Health Organization (WHO) criteria, hospitalized in the Department of Endocrinology, Diabetology and Internal Diseases in Olsztyn between 2015 and 2020. Patient data were obtained from the analysis of hospital medical records. This study was approved by the Bioethical Committee of the School of Medicine, University of Warmia and Mazury in Olsztyn, Poland. The inclusion criteria for the study were DM1 lasting at least a year, AITD diagnosed at least a year prior on the basis of a positive antithyroglobulin antibodies (ATA) titer and ultrasound image of the thyroid gland, and age above 18 years. The exclusion criteria are presented in Figure 1.

The control group (DM1noAITD) consisted of patients without AITD appropriately matched for age, body mass index (BMI) and DM1 duration. Physical examination was performed on both examined groups of patients, taking into account anthropometric measurements and measurements of blood pressure (BP). BP was measured twice using the Korotkoff method in a sitting position after a 10-min rest. Hypertension was diagnosed if systolic blood pressure (SBP) was ≥140 mmHg and diastolic blood pressure (DBP) was ≥90 mmHg, or the patient had already been treated with antihypertensive drugs. All metabolic risk factors for DM such as glycated hemoglobin (HbA1c), lipids and thyroid metabolism were evaluated. Biochemical and hormonal concentration tests were performed at the Laboratory of the Provincial Hospital in Olsztyn according to the standard analytical procedures and quality assurance protocols in force. Venous blood was collected using vacuum/suction sets from the basilic vein, in the morning, while patients were fasting after a 12-h break from eating and drinking. Immediately after transfer to the tubes and clot formation, blood was centrifuged at +4 °C. Its biochemical and hormonal parameters were evaluated in serum by enzymatic-calorimetric and electrochemiluminescence methods using a Roche chemistry analyzer Cobas 6000/c501. The resulting measures included HbA1c, total cholesterol (TC), high-density lipoprotein (HDL)-cholesterol and low-density lipoprotein (LDL)-cholesterol, triglyceride (TG) and creatinine, with the evaluation of the estimated glomerular filtration rate (eGFR) calculated on the basis of creatinine concentration according to the MDRD formula. Thyroid function was assessed by examining the levels of thyroid hormones—free thyroxine (fT4), free triiodothyronine (fT3)—and the pituitary hormone, thyroid-stimulating hormone (TSH), as well as ATA titers, including antithyroid peroxidase antibodies (aTPO), antithyroglobulin antibodies (aTg) and thyroid-stimulating hormone receptor antibodies (TRAb). Patients also underwent thyroid gland ultrasounds. DR was diagnosed and classified on the basis of direct and indirect ophthalmoscopy of the fundus after pupil dilatation with 1% tropicamide, performed by an ophthalmologist. DR status was assessed on the basis of the criteria of the International Clinical Classification for Diabetic Retinopathy, as no DR, non-proliferative diabetic retinopathy (NPDR) or proliferative diabetic retinopathy (PDR) [10,11]. 

### 2.2. Statistical Evaluation of Results

The mean and standard deviation (mean ± SD) were calculated for all tested parameters. Qualitative data are presented as structure indicators (%). The assessment of the normality of the distribution of obtained results was based on the Shapiro–Wilk test. The Student’s t-test was used to assess the statistical significance of differences between the examined groups. If the variables did not meet the normality criteria, the Mann–Whitney U test was performed. The relationship between AITD and DR incidence was assessed by logistic regression as an odds ratio (OR) at a confidence interval (CI) of 95%. The significance level of all statistical analyses was set at α = 0.05. The statistical analysis was carried out using the Statistica 13.0 PL program for Windows.

## 3. Results

In total, the hospital records of 200 patients aged 36 ± 12 years were analyzed, including 70 (35%) men and 130 (65%) women. Thirteen patients were excluded because they did not match the inclusion criteria. The mean duration of DM1 in the whole group was 13 ± 10 years, and the percentage of HbA1c was 9.0 ± 2% (74.9 ± 10.1 mmol/mol). The study group (DM1AITD) consisted of 89 patients diagnosed with DM1 and AITD, aged 35 ± 13 years, 81 (91%) women and 8 (9%) men. The control group consisted of 111 patients with DM1 without AITD matched for age and BMI, aged 36 ± 12 years, 49 (44%) women and 62 (56%) men. The average duration of DM1 in the study group was 12 ± 10 years, and in the control group 13 ± 9 years. In the study group, creatinine concentration was significantly lower than in the control group. DM metabolic risk factors differed significantly, with lower SBP, HbA1c percentage and TG concentration, and higher HDL-cholesterol concentration in the study group than in the control group. Anti-TPO and anti-Tg levels were significantly higher in the study group than in the control group; TRAb were not marked in patients. All patients in DM1AITD had levothyroxine substitution therapy. The data obtained are presented in Table 1.

In the study group, 13 (14.61%) patients had NPDR and 4 (4.49%) PDR (Figure 2). 

Logistic regression analysis was used to determine the relationship between the effects of AITD in DM1 patients and the occurrence of DR. There was a significantly lower chance of NPDR among DM1AITD patients than in the control group. The relationships are presented in Table 2 and Table 3.

## 4. Discussion

It was first stated in the NHANES III study that thyroid disease was more common in patients with DM than in the non-DM population. Examining patients with DM, Perros et al. showed a general incidence of thyroid disease at 13.4% in this group, with women with DM1 (31.4%) noted most often, and men with type 2 DM most rarely (6.9%) [12]. This fact was confirmed by our own research, where in the study group with DM1 and AITD as many as 91% were women. Long-term hyperglycemia leads to the development of chronic macrovascular and microvascular complications, leading to the damage, dysfunction and insufficiency of various organs, especially the eyes and kidneys as well as the cardiovascular system [13]. DR is the most serious eye complication caused by DM and leads to blindness. It develops slowly and without symptoms for a long time [14]. DR is defined as chronic damage to the eye caused by changes in the retinal capillaries and retina, which to varying degrees and over varying lengths of time develops in almost all DM patients. Clinically, DR is divided into an early stage —NPDR—and a more advanced stage, PDR [15,16]. The development of DR occurs in stages, which are determined on the basis of an ophthalmologic examination of the fundus. The basic change in the fundus is microangiopathy leading to the development of neovascularization. The final stage of the clinical manifestation of DR is the loss of vision. DR is responsible for blindness in 80% of patients with DM [17]. The recognized determinants of DR development are disease duration, which is also the strongest prognostic factor for DR development and progression, DM metabolic imbalance—intensive treatment reduces the risk of DR development and progression in DM1 patients—hypertension, lipid metabolism disorders, diabetic kidney disease, pregnancy in women with DM and methods of DM treatment [18]. The impact of these factors is additive, which is why both micro- and macroangiopathy complications are diagnosed most often in patients with a long history of the disease and who are chronically poorly metabolically balanced. In the analyzed group of 200 patients with DM1, the average duration of DM1 was 13 ± 10 years. The longest duration of the disease was 42 years. The interview data show that all patients were diagnosed with DM1 and were treated with insulin from the beginning. The HbA1c percentage in the study group was 8.3% ± 2% (67.2 ± 10.1 mmol/mol), and in the control group 9.5% ± 2% (80.3 ± 10.1 mmol/mol). Earlier studies showed a significant increase in the incidence of DR in cases with an increased HbA1c percentage [19,20,21]. Among DM1 patients intensively treated with insulin, a 76% reduction in the risk of developing DR and a 54% delay in its progression were demonstrated [22]. The common conclusion of numerous studies performed in patients with DM was the fact that intensive glycemic control starting at the diagnosis of DM prevents the occurrence of DR and delays its progression, and the recommended percentage of HbA1c is <7% [23]. In our study, patients diagnosed with AITD had a significantly lower percentage of HbA1c than the control group. It should be emphasized that the average HbA1c value in the whole evaluated group was 9.0% ± 2% (74.9 ± 10.1 mmol/mol). However, the impact of metabolic control on the development of complications in this population cannot be assessed due to the retrospective nature of the observation and the lack of data on DM metabolic control over many years of disease duration. Furthermore, the abnormalities in lipid metabolism in the assessed group of patients are noteworthy as well. A significantly lower TG concentration and higher HDL-cholesterol concentration were found in the study group than in the control group. It is known that low levels of this lipoprotein contribute to the development of atherosclerotic lesions in patients with DM1 [24]. In turn, elevated HDL-cholesterol levels above 60 mg/dL may play a protective role in the development of chronic DM complications, both microangiopathological and cardiovascular. HDL-cholesterol concentration of ≥60 mg/dl is associated with a lower risk of developing DR [25]. It is assumed that thyroid dysfunction may have a significant impact on patients’ lipid profile [26]. In the study group, all patients were treated with levothyroxine. It would be interesting to analyze whether AITD and levothyroxine preparations have a positive effect on the lipid profile of DM1 patients and thus reduce the risk of developing DR. In people with euthyroid disease and DM, the levels of triiodothyronine (T3), TSH and thyrotropin-releasing hormone (TRH) are subject to glycemic changes in two major areas. Glycemia is involved in the regulation of hypothalamic-pituitary-thyroid axis feedback and TSH release from the hypothalamus; it also affects the conversion of thyroxine (T4) to T3 in peripheral tissues. The nocturnal TSH peak and the TSH response to TRH are impaired in DM patients [27,28,29]. Reduced T3 levels which normalize with improved glycemic control have been seen in patients with decompensated DM. Coexisting DM may also affect the efficacy of hypothyroid substitution treatment with levothyroxine [30,31]. Unrecognized thyroid dysfunction can also affect metabolic control in DM. Interactions between thyroid hormones and mechanisms controlling appetite, energy expenditure and insulin sensitivity are also important. A better understanding of this multi-faceted relationship between DM and thyroid disease can help optimize the treatment of DM patients [32]. The treatment of thyroid dysfunction in DM patients has a beneficial effect on glycemic control, reduces cardiovascular risk and improves patients’ overall well-being [2]. To date, little research has been done into the effects of AITD on the development and course of DR in patients with DM1. A literature analysis of the occurrence of DR in patients with DM1, however, shows a significant impact of thyroid disease on the development of DR. By examining the DM1 patient population in Brazil, Rodacki M. et al. proved that TSH levels at 0.4–2.5 mU/L are associated with a lower risk of DR and renal failure in people with DM1, regardless of glycemic control and the duration of the diseases [33]. Quite unexpectedly, Rogowicz-Frontczak A. et al. showed in their study that patients with type 1 DM and positive aTPO, aTg or TRAb antibodies develop microangiopathy to a lesser extent when compared with a group without AITD. They were the first to indicate that DR was less common among patients with type 1 DM and AITD [34]. Our study confirmed this surprising dependence in a larger study group. In our own study, in 80.9% of patients in the study group and 54.95% in the control group no changes to the fundus were demonstrated. Overall, 19.1% of patients in the study group and 45.05% from the control group were diagnosed with DR with varying degrees of severity. NPDR exponents were found in 14.61% of patients with DM1 and AITD and 39.64% of patients in the DM1noAITD group. PDR was diagnosed in the remaining patients. Krolewski et al. found that PDR usually appears after about 10 years of illness, and then occurs at a constant frequency of three cases per 100 per year. After 40 years of illness, the cumulative risk of PDR is 62% [35]. In our own study, despite the long duration of DM, only 4.49% of the patients from the study group and 5.41% from the control group showed PDR exponents. This is most likely related to a significant improvement in diabetes care in recent years, improvement in patients’ metabolic control, the introduction of systematic screening and greater access to specialist ophthalmic care [36,37]. Probably, the genetic profile of patients with autoimmune polyglandular syndrome has a protective effect on the development of DR, despite the higher risk of overt thyroid disease and other organ-specific disorders [38]. The effect of thyroid preparations on DR was analyzed in a 2-year-long study by Schneider et al. conducted on 19 patients. Twelve out of 19 cases showed improvement in thyroid preparations with a reduction in hemorrhage and exudate, but these results have not been confirmed in a larger population [39]. Our own research shows that among patients with DM1 diagnosed with AITD there is a significantly lower chance of the occurrence of NPDR than in the control group. The exact mechanism of this effect is unknown and requires further investigation. In the course of DM, microvascular complications can lead to damage to any organ. However, their earliest and most common clinical manifestation concerns the eye. The first exponents of NPDR may appear after about 4–5 years of the disease [22,40]. Klein et al. showed that in the fifth year of DM1, NPDR occurs in one in 100 patients, but that after 15 years of disease duration, the cumulative NPDR risk in this population approaches 100% [41]. Chronic complications such as micro- and macroangiopathy, despite significant progress in the diagnosis and treatment of DM, are still a significant clinical and social problem. Our study is the first to note other interesting relationships such as the protective effect of thyroid disease and levothyroxine substitution treatment on the development and course of DR in patients with DM. This is a unique discovery that may help to prevent vision loss and improve the quality of life for patients with DM. However, our study has a few limitations. First, this study is cross-sectional and a causal relationship between AITD in patients with DM1 and DR cannot be established; thus, lower DR among DM1 patients must be interpreted with caution. DM metabolic control parameters differed significantly, with lower systolic arterial pressure, HbA1c percentage and TG concentration, and higher HDL-cholesterol concentration. It is difficult to determine whether the outcome followed exposure in time or exposure resulted from the outcome. Further studies with longitudinal prospective design and the presence of optimal BP, lipid profile and a glycemic control matched group are needed to shed more light on the potential relationship between AITD and DR and to assess the relationship between various variables including diabetes control. It is important to state that AITD screening measures are needed. The second limitation of this study was HbA1c measurement. The presence of many clinically silent hemoglobin variants can cause deviation in the HbA1c results, leading to falsely high or low values [42,43]. Therefore, the accuracy of the results and validity of the HbA1c interpretation cannot be assured. Finally, gender differences in the risk of AITD among DM1 patients need further investigation. Using a larger sample size is needed since univariate analysis showed such gender differences, with a higher risk among females. A community-based study among DM1 women will be of great value given that women are more often affected by AITD than men. Although the conclusions require confirmation with a larger group of patients, it seems likely that the presence of AITD in patients with type 1 DM may have an influence on metabolic parameters and the risk of diabetic retinopathy as observed in this study. However, further research requires the identification of factors conditioning DR development and progression as well as factors playing a protective role in this respect.

## 5. Conclusions

Patients with DM1 and AITD were better metabolically balanced, as evidenced by significantly lower SBP, percentage of HbA1c and TG, as well as significantly higher HDL-cholesterol in this group. The study showed a significantly smaller chance of NPDR development among patients with DM1 and AITD.

## Figures and Tables

**Figure 1 medicina-56-00255-f001:**
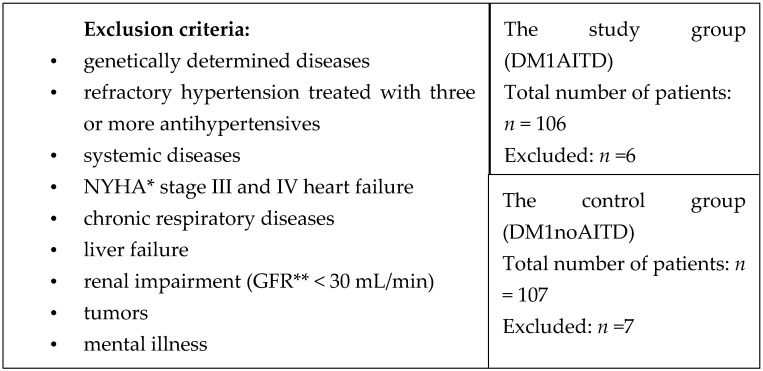
Exclusion criteria. The total number of patients with DM1 and excluded patients. * New York Heart Association Classification, ** glomerular filtration rate. Abbreviations: DM1, diabetes mellitus type 1.

**Figure 2 medicina-56-00255-f002:**
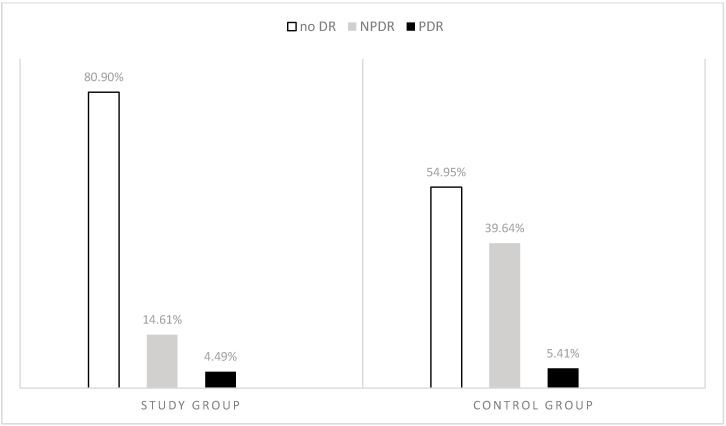
Occurrence of DR, *p* < 0.01 *. Study group: DM1 + AITD. Control group: DM1noAITD. Study group vs control group. * chi-square test. Abbreviations: DR, diabetic retinopathy; NPDR, non-proliferative diabetic retinopathy; PDR, proliferative diabetic retinopathy.

**Table 1 medicina-56-00255-t001:** Characteristics of both groups, parameters of DM metabolic control and thyroid function assessment. Study group: DM1 + AITD. Control group: DM1noAITD. *p* < 0.05 study group vs. control group. * Mann-Whitney U test.

Parameters	Study Group	Control Group	*p*
Number of patients (*n*)	89	111	
Women/Men (*n*)	8–81	49/62	
Age (years)	35 ± 13	36 ± 12	0.31
Duration of DM1 (years)	12 ± 10	13 ± 9	0.15
BMI (kg/m²)	24.18 ± 4	23.27 ± 3	0.16 *
Creatinine (mg/dL)	0.75 ± 0.3	0.87 ± 0.5	<0.01
eGFR (mL/min/1.73 m²)	94.18 ± 30.1	100.06 ± 26.6	0.38 *
SBP (mmHg)	116 ± 12	118 ± 12	0.22 *
DBP (mmHg)	76 ± 10	78 ± 9	0.04 *
HbA1c (%)	8.3 ± 2	9.5 ± 2	<0.01
HbA1c (mmol/mol)	67.2 ± 10.1	80.3 ± 10.1	<0.01
TC (mg/dL)	185 ± 44	180 ± 46	0.47
LDL-c (mg/dL)	108 ± 39	104 ± 35	0.57
HDL-c (mg/dL)	72 ± 18	64 ± 22	<0.01 *
TG (mg/dL)	96 ± 50	120 ± 69	<0.01
TSH (mU/L)	2.57 ± 5.12	1.88 ± 0.89	0.61 *
fT3 (pmol/L)	4.31 ± 0.72	4.43 ± 0.85	0.26 *
fT4 (pmol/L)	16.94 ± 3.86	16.10 ± 2.42	0.18 *
aTPO (IU/mL)	194 ± 180	14.65 ± 7	<0.01
aTg (IU/mL)	193 ± 250	16 ± 12	<0.01

Abbreviations: DM1, diabetes mellitus type 1; eGFR, estimated glomerular filtration rate; SBP, systolic blood pressure; DBP, diastolic blood pressure; HbA1c, glycated hemoglobin; TC, total cholesterol; LDL-c, low-density lipoprotein cholesterol; HDL-c, high-density lipoprotein cholesterol; TG, triglyceride; TSH, thyroid-stimulating hormone; fT3, free triiodothyronine; fT4, free thyroxine; aTPO, antithyroid peroxidase antibodies; aTg, antithyroglobulin antibodies; DM, diabetes mellitus.

**Table 2 medicina-56-00255-t002:** Odds Ratio (OR) of PDR in patients in the study group (DM1 + AITD) and in the control group DM1noAITD. Abbreviations: PDR, proliferative diabetic retinopathy; CI, confidence interval.

	Odds Ratio (95% CI)
Study group	0.5648 (0.1506–2.1177)
Control group	1
*p*	0.39

**Table 3 medicina-56-00255-t003:** Odds Ratio (OR) occurrence of NPDR in patients in the study group (DM1 + AITD) and in the control group DM1noAITD. Abbreviations: NPDR, non-proliferative diabetic retinopathy.

	Odds Ratio (95% CI)
Study group	0.2503 (0.1229–0.5097)
Control group	1
*p*	<0.01

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
