# Peer review of "Autoimmune Thyroid Disease is Associated with a Lower Prevalence of Diabetic Retinopathy in Patients with Type 1 Diabetic Mellitus"

_medicina, 2020, doi:10.3390/medicina56060255_

Round 1

Reviewer 1 Report

This is an interesting analysis investigating whether the presence of Autoimmune thyroid disease in patients with DM1 has an influence of metabolic parameters and risk of diabetic retinopathy. 

In general it is well written and easy to understand. 

 Title: It is not common to ask a question in a title, it could be more catchy to use a title such as “Presence of AITD is associated with a lower prevalence of diabetic retinopathy in individuals with Type 1 Diabetes mellitus”.

Abstract: 

  • It is kind of confusing to call those without AITD control group. There could be considered another nomenclature as DM1AITD and DM1noAITD.
  • The sentence in line 22 "All patients were diagnosed with..." can be deleted.
  • The wording diabetes metabolic control is not usual. I would suggest to substitute it to metabolic risk factors as HbA1c, lipids and blood pressure.
  • Results: the primary objective is not mentioned in the results section but only in the Conclusions 
  • Conclusions:
  • Add your significant findings on cardiovascular risk factors as well
  • Do not abbreviate NDPR at its first appearance

Introduction:

  • Line 32: DR is a microvascular and not neurovascular complication please amend.
  • The introduction is lacking of details whereas these information are mostly given in the beginning of the discussion. This should be changed.
  • Please add the primary objective of your analysis in the end of the introduction part

Aim:

  • I would change occurrence to presence as you have no timely data presented

Material and Methods:

  • Add the study design in the beginning (retrospective, cross-sectional, non interventional)
  • Line: 52: eGFR kidney failure: please change to renal impairment (GFR<30ml/min)
  • Line 59: diabetes metabolic control – see above
  • Line 62: I hope you did not gain venous blood from the tracheal vein… Please declare
  • Line 71: Please add thyroid specific antibodies which were measured

Results:

  • Please also indicate HbA1c in mmol/mol
  • Line 97 : metabolic control parameters – see above

Table 1:

  • Please add which type of AITD the people were suffering from (Graves vs. Hashimoto vs. both)
  • Do you have information of whether patients had substitution therapy and which dose, if yes please indicate

Table 3. Please add TRAK

Line 133. Does this sentence belong to the results section or the table?

Discussion:

  • Start your discussion with the main findings of your analysis. Skip the general information and put it in the background section.
  • Please add important limitations of your analysis
    • Cross sectional design
    • No data about influence of glycemic control on RP or metabolic risk factors

The literature used throughout the manuscript seems to be not up to date (e.g. ADA Guidelines from 2011 are used); please check again and  amend.

Author Response

Response to Reviewer 1 Comments

We thank you very much for yours useful suggestions. Our analysis were one of the first to indicate that diabetic retinopathy was less common among patients with type 1 DM and AITD. Although conclusions require confirmation with a larger group of patients, it seems likely that the presence of AITD in patients with type 1 DM may have an influence on metabolic parameters and risk of diabetic retinopathy observed in this study.

 Title: It is not common to ask a question in a title, it could be more catchy to use a title such as “Presence of AITD is associated with a lower prevalence of diabetic retinopathy in individuals with Type 1 Diabetes mellitus”.

Response 1:

We understand that it is not common to ask this question. We have changed a title according to your suggestion : “Presence of autoimmune thyroid disease is associated with a lower prevalence of diabetic retinopathy in patients with type 1 diabetes mellitus”.

Abstract: 

  • It is kind of confusing to call those without AITD control group. There could be considered another nomenclature as DM1AITD and DM1noAITD.
  • The sentence in line 22 "All patients were diagnosed with..." can be deleted.
  • The wording diabetes metabolic control is not usual. I would suggest to substitute it to metabolic risk factors as HbA1c, lipids and blood pressure.
  • Results: the primary objective is not mentioned in the results section but only in the Conclusions 
  • Conclusions:
  • Add your significant findings on cardiovascular risk factors as well
  • Do not abbreviate NDPR at its first appearance

Response 2:

It was corrected according to your suggestions.

Introduction:

  • Line 32: DR is a microvascular and not neurovascular complication please amend.
  • The introduction is lacking of details whereas these information are mostly given in the beginning of the discussion. This should be changed.
  • Please add the primary objective of your analysis in the end of the introduction part

Response 3:

It was corrected according to your suggestions.

Aim:

  • I would change occurrence to presence as you have no timely data presented

Response 4:

It was corrected according to your suggestions.

Material and Methods:

  • Add the study design in the beginning (retrospective, cross-sectional, non interventional)
  • Line: 52: eGFR kidney failure: please change to renal impairment (GFR<30ml/min)
  • Line 59: diabetes metabolic control – see above
  • Line 62: I hope you did not gain venous blood from the tracheal vein… Please declare
  • Line 71: Please add thyroid specific antibodies which were measured

Response 5:

It was corrected according to your suggestions.

Results:

  • Please also indicate HbA1c in mmol/mol
  • Line 97 : metabolic control parameters – see above

Table 1:

  • Please add which type of AITD the people were suffering from (Graves vs. Hashimoto vs. both)
  • Do you have information of whether patients had substitution therapy and which dose, if yes please indicate

Table 3. Please add TRAK

Line 133. Does this sentence belong to the results section or the table?

Response 6:

It was corrected according to your suggestions. Patients had only Hashimoto’s thyroiditis. TRAb were not found in both groups of patients, therefore not included. Line 133. This sentence belongs to the results section.

Discussion:

  • Start your discussion with the main findings of your analysis. Skip the general information and put it in the background section.
  • Please add important limitations of your analysis
    • Cross sectional design
    • No data about influence of glycemic control on RP or metabolic risk factors

The literature used throughout the manuscript seems to be not up to date (e.g. ADA Guidelines from 2011 are used); please check again and  amend.

Response 7:

It was corrected according to your suggestions.

Reviewer 2 Report

It is a descriptive and retrospective study of the casuistry of a group. The study is of limited clinical interest.

The control group consisted of patients without AITD appropriately matched for age and BMI BUT NOT IN DM1 DURATION OR METABOLIC CONTROL. The duration ranges of type 1 diabetes, BP and glycemic control are essential for the onset of chronic diabetes complications. These two factors must be controlled in the statistical study since the control group is not mapped on these factors.

The groups are not comparable in these primary factors of what it is intended to study.

In the study group there are more women, DM metabolic control parameters differed significantly, with lower systolic arterial pressure, HbA1c percentage and TG concentration, and higher HDL-cholesterol concentration.

The statistical study must adjust the factors in which the two groups are not homogeneous (metabolic control)

The total number of patients with DM1 and those who have been excluded should be included in a figure (indicating the exclusion criteria).

Tables 1, 2 and 3 can be joined

Statistical evaluation of results is usually in material and methods rather than results.

You conclude that Patients with DM1 and AITD were better metabolically balanced, as evidenced by a significantly lower percentage of HbA1c and TG, as well as significantly higher HDL-cholesterol in this group. In order to demonstrate this claim, they should compare metabolic control before and after the appearance of AITD.

Author Response

Response to Reviewer 2 Comments

It is a descriptive and retrospective study of the casuistry of a group. The study is of limited clinical interest.

Response 1: Thank you very much for yours useful and valuable suggestions. Our analysis were one of the first to indicate that diabetic retinopathy was less common among patients with type 1 DM and AITD. Although conclusions requires confirmation with a larger group of patients, it seems likely that the presence of AITD in patients with type 1 DM may has an influence of risk of diabetic retinopathy observed in this study.

The control group consisted of patients without AITD appropriately matched for age and BMI BUT NOT IN DM1 DURATION OR METABOLIC CONTROL. The duration ranges of type 1 diabetes, BP and glycemic control are essential for the onset of chronic diabetes complications. These two factors must be controlled in the statistical study since the control group is not mapped on these factors.

The groups are not comparable in these primary factors of what it is intended to study.

In the study group there are more women, DM metabolic control parameters differed significantly, with lower systolic arterial pressure, HbA1c percentage and TG concentration, and higher HDL-cholesterol concentration.

The statistical study must adjust the factors in which the two groups are not homogeneous (metabolic control)

Response 2: Thank you very much for yours useful and valuable suggestions. We added this important limitations of our analysis.

The total number of patients with DM1 and those who have been excluded should be included in a figure (indicating the exclusion criteria).

Response 2: It was corrected according to your suggestions.

Tables 1, 2 and 3 can be joined

Response 3: It was corrected according to your suggestions.

Statistical evaluation of results is usually in material and methods rather than results.

Response 4: It was corrected according to your suggestions.

You conclude that Patients with DM1 and AITD were better metabolically balanced, as evidenced by a significantly lower percentage of HbA1c and TG, as well as significantly higher HDL-cholesterol in this group. In order to demonstrate this claim, they should compare metabolic control before and after the appearance of AITD.

Response 2: Thank you very much for yours useful and valuable suggestions. We added this important limitations of our analysis.

Round 2

Reviewer 2 Report

I consider that the article is ready for publication with the changes made